# Regularized Weighted Low Rank Approximation

**Frank Ban**
UC Berkeley / Google
fban@berkeley.edu

**David Woodruff**
Carnegie Mellon University
dwoodruf@cs.cmu.edu

**Qiuyi (Richard) Zhang**
UC Berkeley / Google
qiuyi@berkeley.edu

## Abstract

The classical low rank approximation problem is to find a rank $k$ matrix $UV$ (where $U$ has $k$ columns and $V$ has $k$ rows) that minimizes the Frobenius norm of $A - UV$. Although this problem can be solved efficiently, we study an NP-hard variant of this problem that involves weights and regularization. A previous paper of [Razenshteyn et al. '16] derived a polynomial time algorithm for weighted low rank approximation with constant rank. We derive provably sharper guarantees for the regularized version by obtaining parameterized complexity bounds in terms of the statistical dimension rather than the rank, allowing for a rank-independent runtime that can be significantly faster. Our improvement comes from applying sharper matrix concentration bounds, using a novel conditioning technique, and proving structural theorems for regularized low rank problems.

## 1 Introduction

In the weighted low rank approximation problem, one is given a matrix $M \in \mathbb{R}^{n \times d}$, a weight matrix $W \in \mathbb{R}^{n \times d}$, and an integer parameter $k$, and would like to find factors $U \in \mathbb{R}^{n \times k}$ and $V \in \mathbb{R}^{k \times d}$ so as to minimize

$$\|W \circ (M - U \cdot V)\|_F^2 = \sum_{i=1}^{n} \sum_{j=1}^{d} W_{i,j}^2 (M_{i,j} - \langle U_{i,*}, V_{*,j} \rangle)^2,$$

where $U_{i,*}$ denotes the $i$-th row of $U$ and $V_{*,j}$ denotes the $j$-th column of $V$. W.l.o.g., we assume $n \geq d$. This is a weighted version of the classical low rank approximation problem, which is a special case when $W_{i,j} = 1$ for all $i$ and $j$. One often considers the approximate version of this problem, for which one is given an approximation parameter $\varepsilon \in (0, 1)$ and would like to find $U \in \mathbb{R}^{n \times k}$ and $V \in \mathbb{R}^{k \times d}$ so that

$$\|W \circ (M - U \cdot V)\|_F^2 \leq (1 + \varepsilon) \min_{U' \in \mathbb{R}^{n \times k}, V' \in \mathbb{R}^{k \times d}} \|W \circ (M - U' \cdot V')\|_F^2. \tag{1}$$

Weighted low rank approximation extends the classical low rank approximation problem in many ways. While in principal component analysis, one typically first subtracts off the mean to make the matrix $M$ have mean 0, this does not fix the problem of differing variances. Indeed, imagine one of the columns of $M$ has much larger variance than the others. Then in classical low rank approximation with $k = 1$, it could suffice to simply fit this single high variance column and ignore the remaining entries of $M$. Weighted low rank approximation, on the other hand, can correct for this by re-weighting each of the entries of $M$ to give them similar variance; this allows for the low rank approximation $U \cdot V$ to capture more of the remaining data. This technique is often used in gene expression analysis and co-occurrence matrices; we refer the reader to [SJ03] and the Wikipedia entry on weighted low rank approximation[1]. The well-studied problem of *matrix completion* is

also a special case of weighted low rank approximation, where the entries $W_{i,j}$ are binary, and has numerous applications in recommendation systems and other settings with missing data.

Unlike its classical variant, weighted low rank approximation is known to be NP-hard [GG11]. Classical low rank approximation can be solved quickly via the singular value decomposition, which is often sped up with sketching techniques [Woo14, PW15, TYUC17]. However, in the weighted setting, under a standard complexity-theoretic assumption known as the Random Exponential Time Hypothesis (see, e.g., Assumption 1.3 in [RSW16] for a discussion), there is a fixed constant $\varepsilon_0 \in (0,1)$ for which any algorithm achieving (1) with constant probability and for $\varepsilon = \varepsilon_0$, and even for $k = 1$, requires $2^{\Omega(r)}$ time, where $r$ is the number of distinct columns of the weight matrix $W$. Furthermore, as shown in Theorem 1.4 of [RSW16], this holds even if $W$ has both at most $r$ distinct rows and $r$ distinct columns.

Despite the above hardness, in a number of applications the parameter $r$ may be small. Indeed, in a matrix in which the rows correspond to users and the columns correspond to ratings of a movie, such as in the Netflix matrix, one may have a small number of categories of movies. In this case, one may want to use the same column in $W$ for each movie in the same category. It may thus make sense to renormalize user ratings based on the category of movies being watched. Note that any number of distinct rows of $W$ is possible here, as different users may have completely different ratings, but there is just one distinct column of $W$ per category of movie. In some settings one may simultaneously have a small number of distinct rows and a small number of distinct columns. This may occur if say, the users are also categorized into a small number of groups. For example, the users may be grouped by age and one may want to weight ratings of different categories of movies based on age. That is, maybe cartoon ratings of younger users should be given higher weight, while historical films rated by older users should be given higher weight.

Motivated by such applications when $r$ is small, [RSW16] propose several *parameterized complexity algorithms*. They show that in the case that $W$ has at most $r$ distinct rows and $r$ distinct columns, there is an algorithm solving (1) in $2^{O(k^2 r/\varepsilon)}\text{poly}(n)$ time. If $W$ has at most $r$ distinct columns but any number of distinct rows, there is an algorithm achieving (1) in $2^{O(k^2 r^2/\varepsilon)}\text{poly}(n)$ time. Note that these bounds imply that for constant $k$ and $\varepsilon$, even if $r$ is as large as $\Theta(\log n)$ in the first case, and $\Theta(\sqrt{\log n})$ in the second case, the corresponding algorithm is polynomial time.

In [RSW16], the authors also consider the case when the rank of the weight matrix $W$ is at most $r$, which includes the $r$ distinct rows and columns, as well as the $r$ distinct column settings above, as special cases. In this case the authors achieve an $n^{O(k^2 r/\varepsilon)}$ time algorithm. Note that this is only polynomial time if $k, r$, and $\varepsilon$ are each fixed constants, and unlike the algorithms for the other two settings, this algorithm is not fixed parameter tractable, meaning its running time cannot be written as $f(k, r, 1/\varepsilon) \cdot \text{poly}(nd)$, where $f$ is a function that is independent of $n$ and $d$.

There are also other algorithms for weighted low rank approximation, though they do not have provable guarantees, or require strong assumptions on the input. There are gradient-based approaches of Shpak [Shp90] and alternating minimization approaches of Lu et al. [LPW97, LA03], which were refined and used in practice by Srebro and Jaakkola [SJ03]. However, neither of these has provable gurantees. While there is some work that has provable guarantees, it makes incoherence assumptions on the low rank factors of $M$, as well as assumptions that the weight matrix $W$ is spectrally close to the all ones matrix [LLR16] and that there are no 0 weights.

## 1.1 Our Contributions

The only algorithms with provable guarantees that do not make assumptions on the inputs are slow, and inherently so given the above hardness results. Motivated by this and the widespread use of regularization in machine learning, we propose to look at *provable guarantees for regularized weighted low rank approximation*. In one version of this problem, where the parameter $r$ corresponds to the rank of the weight matrix $W$, we are given a matrix $M \in \mathbb{R}^{n \times d}$, a weight matrix $W \in \mathbb{R}^{n \times d}$ with rank $r$, and a target integer $k > 0$, and we consider the problem

$$\min_{U \in \mathbb{R}^{n \times k}, V \in \mathbb{R}^{k \times d}} \|W \circ (UV - M)\|_F^2 + \lambda\|U\|_F^2 + \lambda\|V\|_F^2$$

Let $U^*, V^*$ minimize $\|W \circ (UV - M)\|_F^2 + \lambda\|U\|_F^2 + \lambda\|V\|_F^2$ and OPT be the minimum value.

Regularization is a common technique to avoid overfitting and to solve an ill-posed problem. It has been applied in the context of weighted low rank approximation [DN11], though so far the only such results known for weighted low rank approximation with regularization are heuristic. In this paper we give the first provable bounds, without any assumptions on the input, on regularized weighted low rank approximation.

Importantly, we show that regularization improves our running times for weighted low rank approximation, as specified below. Intuitively, the complexity of regularized problems depends on the "statistical dimension" or "effective dimension" of the underlying problem, which is often significantly smaller than the number of parameters in the regularized setting.

Let $U^*$ and $V^*$ denote the optimal low-rank matrix approximation factors, $D_{W_{i,:}}$ denote the diagonal matrix with the $i$-th row of $W$ along the diagonal, and $D_{W_{:,j}}$ denote the diagonal matrix with the $j$-th column of $W$ along the diagonal.

**Improving the Exponent:** We first show how to improve the $n^{O(k^2 r/\varepsilon)}$ time algorithm of [RSW16] to a running time of $n^{O((s+\log(1/\varepsilon))rk/\varepsilon)}$. Here $s$ is defined to be the maximum statistical dimension of $V^* D_{W_{i,:}}$ and $D_{W_{:,j}} U^*$, over all $i = 1, \ldots, n$, and $j = 1, \ldots, d$, where the statistical dimension of a matrix $M$ is:

**Definition 1.** *Let $\mathtt{sd}_\lambda(M) = \sum_i 1/(1 + \lambda/\sigma_i^2)$ denote the statistical dimension of $M$ with regularizing weight $\lambda$ (here $\sigma_i$ are the singular values of $M$).*

Note that this maximum value $s$ is always at most $k$ and for any $s \geq \log(1/\varepsilon)$, our bound directly improves upon the previous time bound. Our improvement requires us to sketch matrices with k columns down to $s/\varepsilon$ rows where $s/\varepsilon$ is potentially smaller than $k$. This is particularly interesting since most previous provable sketching results for low rank approximation cannot have sketch sizes that are smaller than the rank, as following the standard analysis would lead to solving a regression problem on a matrix with fewer rows than columns.

Thus, we introduce the notion of an upper and lower distortion factor ($K_S$ and $\kappa_S$ below) and show that the lower distortion factor will satisfy tail bounds only on a smaller-rank subspace of size $s/\varepsilon$, which can be smaller than k. Directly following the analysis of [RSW16] will cause the lower distortion factor to be infinite. The upper distortion factor also satisfies tail bounds via a more powerful matrix concentration result not used previously. Furthermore, we apply a novel conditioning technique that conditions on the product of the upper and lower distortion factors on separate subspaces, whereas previous work only conditions on the condition number of a specific subspace.

We next considerably strengthen the above result by showing an $n^{O(r^2(s+\log(1/\varepsilon))^2/\varepsilon^2))}$ time algorithm. This shows that the rank $k$ need not be in the exponent of the algorithm at all! We do this via a novel projection argument in the objective (sketching on the right), which was not done in [RSW16] and also improves a previous result for the classical setting in [ACW17]. To gain some perspective on this result, suppose $\varepsilon$ is a large constant, close to 1, and $r$ is a small constant. Then our algorithm runs in $n^{O(s^2)}$ time as opposed to the algorithm of [RSW16] which runs in $n^{O(k^2)}$ time. We stress in a number of applications, the effective dimension $s$ may be a very small constant, close to 1, even though the rank parameter $k$ can be considerably larger. This occurs, for example, if there is a single dominant singular value, or if the singular values are geometrically decaying. Concretely, it is realistic that $k$ could be $\Theta(\log n)$, while $s = \Theta(1)$, in which case our algorithm is the first polynomial time algorithm for this setting.

**Improving the Base:** We can further optimize by removing our dependence on $n$ in the base. The *non-negative rank* of a $n \times d$ matrix $A$ is defined to be the least $r$ such that there exist factors $U \in \mathbb{R}^{n \times r}$ and $V \in \mathbb{R}^{r \times d}$ where $A = U \cdot V$ and both $U$ and $V$ have non-negative entries. By applying a novel rounding procedure, if in addition the non-negative rank of $W$ is at most $r'$, then we can obtain a fixed-parameter tractable algorithm running in time $2^{r' r^2 (s+\log(1/\varepsilon))^2/\varepsilon^2)} \text{poly}(n)$. Note that $r \leq r'$, where $r$ is the rank of $W$. Note also that if $W$ has at most $r$ distinct rows or columns, then its non-negative rank is also at most $r$ since we can replace the entries of $W$ with their absolute values without changing the objective function, while still preserving the property of at most $r$ distinct rows and/or columns. Consequently, we significantly improve the algorithms for a small number of distinct rows and/or columns of [RSW16], as our exponent is *independent* of $k$.

Thus, even if $k = \Theta(n)$ but the statistical dimension $s = O(\sqrt{\log n})$, for constant $r'$ and $\varepsilon$ our algorithm is polynomial time, while the best previous algorithm would be exponential time.

We also give ways, other than non-negative rank, for improving the running time. Supposing that the rank of $W$ is $r$ again, we apply iterative techniques in linear system solving like Richardson's Iteration and preconditioning to further improve the running time. We are able to show that instead of an $n^{\mathrm{poly}(rs/\varepsilon)}$ time algorithm, we are able to obtain algorithms that have running time roughly $(\sigma^2/\lambda)^{\mathrm{poly}(rs/\varepsilon)}\mathrm{poly}(n)$ or $(u_W/l_W)^{\mathrm{poly}(rs/\varepsilon)}\mathrm{poly}(n)$, where $\sigma^2$ is defined to be the maximum singular value of $V^* D_{W_{i,:}}$ and $D_{W_{:,j}} U^*$, over all $i = 1, \ldots, n$, and $j = 1, \ldots, d$, while $u_W$ is defined to be the maximum absolute value of an entry of $W$ and $l_W$ the minimum absolute value of an entry. In a number of settings one may have $\sigma^2/\lambda = O(1)$ or $u_W/l_W = O(1)$ in which case we again obtain fixed parameter tractable algorithms.

**Empirical Evaluation:** Finally, we give an empirical evaluation of our results. While the main goal of our work is to obtain the first algorithms with provable guarantees for regularized weighted low rank approximation, we can also use them to guide heuristics in practice. In particular, alternating minimization is a common heuristic for weighted low rank approximation. We consider a sketched version of alternating minimization to speed up each iteration. We show that in the regularized case, the dimension of the sketch can be significantly smaller if the statistical dimension is small, which is consistent with our theoretical results.

## 2    Preliminaries

We let $\|\cdot\|_F$ denote the Frobenius norm of a matrix and let $\circ$ be the elementwise matrix multiplication operator. We denote $x \in [a, b]\, y$ if $ay \leq x \leq by$. For a matrix $M$, let $M_{i,:}$ denote its $i$th row and let $M_{:,j}$ denote its $j$th column. For $v \in \mathbb{R}^n$, let $D_v$ denote the $n \times n$ diagonal matrix with its $i$-th diagonal entry equal to $v_i$. For a matrix $M$ with non-negative $M_{ij}$, let $\mathtt{nnr}(M)$ denote the non-negative rank of $M$. Let $\mathtt{sr}(M) = \|M\|_F^2/\|M\|^2$ denote the stable rank of $M$. Let $\mathcal{D}$ denote a distribution over $r \times n$ matrices; in our setting, there are matrices with entries that are Gaussian random variables with mean 0 and variance $1/r$ (or $r \times n$ CountSketch matrices [Woo14]).

**Definition 2.** *For $S$ sampled from a distribution of matrices $\mathcal{D}$ and a matrix $M$ with $n$ rows, let $c_S(M) \geq 1$ denote the smallest (possibly infinite) number such that $\|SMv\|^2 \in [c_S(M)^{-1}, c_S(M)]\|Mv\|^2$ for all $v$.*

**Definition 3.** *For $S$ sampled from a distribution of matrices $\mathcal{D}$ and a matrix $M$, let $K_S(M) \geq 1$ denote the smallest number such that $\|SMv\|^2 \leq K_S(M)\|Mv\|^2$ for all $v$.*

**Definition 4.** *For $S$ sampled from a distribution of matrices $\mathcal{D}$ and a matrix $M$, let $\kappa_S(M) \leq 1$ denote the largest number such that $\|SMv\|^2 \geq \kappa_S(M)\|Mv\|^2$ for all $v$.*

Note that by definition, $c_s(M)$ equals the max of $K_S(M)$ and $\frac{1}{\kappa_S(M)}$. We define the condition number of a matrix $A$ to be $c_A = K_A(I)/\kappa_A(I)$.

### 2.1    Previous Techniques

Building upon the initial framework established in [RSW16], we apply a polynomial system solver to solve weighted regularized LRA to high accuracy. By applying standard sketching guarantees, $v$ can be made a polynomial function of $k, 1/\varepsilon, r$ that is independent of $n$.

**Theorem 1** ([Ren92a][Ren92b][BPR96])**.** *Given a real polynomial system $P(x_1, x_2, ..., x_v)$ having $v$ variables and $m$ polynomial constraints $f_i(x_1, ..., x_v)\Delta_i 0$, where $\Delta_i \in \{\geq, =, \leq\}$, $d$ is the maximum degree of all polynomials, and $H$ is the maximum bitsize of the coefficients of the polynomials, one can determine if there exists a solution to $P$ in $(md)^{O(v)}poly(H)$ time.*

Intuitively, the addition of regularization requires us to only preserve directions with high spectral weight in order to preserve our low rank approximation well enough. This dimension of the subspace spanned by these important directions is exactly the statistical dimension of the problem, allowing us to sketch to a size less than $k$ that could provably preserve our low rank approximation well enough. In line with this intuition, we use an important lemma from [CNW16]

**Lemma 2.1.** *Let $A, B$ be matrices with $n$ rows and let $S$, sampled from $\mathcal{D}$, have $\ell = \Omega(\frac{1}{\gamma^2}(K + \log(1/\varepsilon)))$ rows and $n$ columns. Then*

$$\mathbf{Pr}\left[\|A^T S^T S B - A^T B\| > \gamma \cdot \sqrt{\left(\|A\|^2 + \|A\|_F^2/K\right)\left(\|B\|^2 + \|B\|_F^2/K\right)}\right] < \varepsilon$$

*In particular, if we choose $K > \Omega(sr(A) + sr(B))$, then we have for some small constant $\gamma'$,*

$$\mathbf{Pr}\left[\|A^T S^T S B - A^T B\| > \gamma' \|A\|\|B\|\right] < \varepsilon$$

## 3 Multiple Regression Sketches

In this section, we prove our main structural theorem which allows us to sketch regression matrices to the size of the statistical dimension of the matrices while maintaining provable guarantees. Specifically, to approximately solve a sum of regression problems, we are able to reduce the dimension of the problem to the maximum statistical dimension of the regression matrices.

**Theorem 2.** *Let $M^{(1)}, \dots, M^{(d)} \in \mathbb{R}^{n \times k}$ and $b^{(1)}, \dots, b^{(d)} \in \mathbb{R}^n$ be column vectors. Let $S \in \mathbb{R}^{\ell \times n}$ be sampled from $\mathcal{D}$ with $\ell = \Theta(\frac{1}{\varepsilon}(s + \log(1/\varepsilon)))$ and $s = \max_i \{sd_\lambda(M^{(i)})\}$.*

*Define $x^{(i)} = \operatorname{argmin}_x \|M^{(i)}x - b^{(i)}\|^2 + \lambda\|x\|^2$ and $y^{(i)} = \operatorname{argmin}_y \|S(M^{(i)}y - b^{(i)})\|^2 + \lambda\|y\|^2$. Then, with constant probability,*

$$\sum_{i=1}^{d} \|M^{(i)}y^{(i)} - b^{(i)}\|^2 + \lambda\|y^{(i)}\|^2 \le (1 + \varepsilon) \cdot \left(\sum_{i=1}^{d} \|M^{(i)}x^{(i)} - b^{(i)}\|^2 + \lambda\|x^{(i)}\|^2\right)$$

We note that a simple union bound would incur a dependence of a factor of $\log(d)$ in the sketching dimension $l$. While this might seem mild at first, the algorithms we consider are exponential in $l$, implying that we would be unable to derive polynomial time algorithms for solving weighted low rank approximation even when the input and weight matrix are both of constant rank. Therefore, we need an average case version of sketching guarantees to hold; however, this is not always the case since $l$ is small and applying Lemma 2.1 naïvely only gives a probability bound. Ultimately, we must condition on the event of a combination of sketching guarantees holding and carefully analyzing the expectation in separate cases.

## 4 Algorithms

In this section, we present a fast algorithm for solving regularized weighted low rank approximation. Our algorithm exploits the structure of low-rank approximation as a sum of regression problems and applies the main structural theorem of our previous section to significantly reduce the number of variables in the optimization process. Note that we can write

$$\|W \circ (UV - A)\|_F^2 = \sum_{i=1}^{n} \|U_{i,:} V D_{W_{i,:}} - A_{i,:} D_{W_{i,:}}\|^2 = \sum_{j=1}^{d} \|D_{W_{:,j}} U V_{:,j} - D_{W_{:,j}} A_{:,j}\|^2$$

### 4.1 Using the Polynomial Solver with Sketching

Now we sample Gaussian sketch matrices $S'$ from $\mathbb{R}^{d \times \Theta(\frac{s}{\varepsilon})\log(1/\varepsilon)}$ and $S''$ from $\mathbb{R}^{\Theta(\frac{s}{\varepsilon})\log(1/\varepsilon) \times n}$. We let $P^{(i)}$ denote $V D_{W_{i,:}} S'$ and $Q^{(j)}$ denote $S'' D_{W_{:,j}} U$.

The matrices $P^{(i)}$ and $Q^{(j)}$ can be encoded using $\Theta(\frac{s + \log(1/\varepsilon)}{\varepsilon})kr$ variables. For fixed $P^{(i)}$ and $Q^{(j)}$ we can define

$$\tilde{U} = \operatorname*{argmin}_{U \in \mathbb{R}^{n \times k}} \sum_{i=1}^{n} \|U_{i,:} P^{(i)} - A_{i,:} D_{W_{i,:}} S'\|^2 + \lambda\|U_{i,:}\|^2$$

and

$$\tilde{V} = \operatorname*{argmin}_{V \in \mathbb{R}^{k \times n}} \sum_{j=1}^{d} \|Q^{(j)} V_{:,j} - S'' D_{W_{:,j}} A_{:,j}\|^2 + \lambda\|V_{:,j}\|^2$$

---

**Algorithm 1** Regularized Weighted Low Rank Approximation

---

**public : procedure** REGWEIGHTEDLOWRANK$(A, W, \lambda, s, k, \varepsilon)$

   Sample Gaussian sketch $S' \in \mathbb{R}^{d \times \Theta(\frac{s}{\varepsilon}) \log(1/\varepsilon)}$ from $\mathcal{D}$

   Sample Gaussian sketch $S'' \in \mathbb{R}^{\Theta(\frac{s}{\varepsilon}) \log(1/\varepsilon) \times n}$ from $\mathcal{D}$.

   Create matrix variables $P^{(i)} \in \mathbb{R}^{k \times \Theta(\frac{s}{\varepsilon}) \log(1/\varepsilon)}, Q^{(j)} \in \mathbb{R}^{k \times \Theta(\frac{s}{\varepsilon}) \log(1/\varepsilon)}$ for $i, j$ from 1 to $r$
   $\triangleright$ Variables used in polynomial system solver

   Use Cramer's Rule to express $\tilde{U}_{i,:} = A_{i,:} D_{W_{i,:}} S' (P^{(i)})^T (P^{(i)} (P^{(i)})^T + \lambda I_k)^{-1}$ as a rational
   function of variables $P^{(i)}$; similarly, $\tilde{V}_{:,j} = ((Q^{(j)})^T Q^{(j)} + \lambda I_k)^{-1} (Q^{(j)})^T S'' D_{W_{:,j}} A_{:,j}$
   $\triangleright \tilde{U}, \tilde{V}$ are now rational function of variables $P, Q$

   Solve $\min_{\tilde{U}, \tilde{V}} \|W \circ (\tilde{U}\tilde{V} - A)\|_F^2 + \lambda \|\tilde{U}\|_F^2 + \lambda \|\tilde{V}\|_F^2$ and apply binary search to find $\tilde{U}, \tilde{V}$
   $\triangleright$ Optimization with polynomial solver of Theorem 1 in variables $P, Q$

   **return** $\tilde{U}, \tilde{V}$

---

to get

$$\tilde{U}_{i,:} = A_{i,:} D_{W_{i,:}} S' (P^{(i)})^T (P^{(i)} (P^{(i)})^T + \lambda I_k)^{-1}$$

and

$$\tilde{V}_{:,j} = ((Q^{(j)})^T Q^{(j)} + \lambda I_k)^{-1} (Q^{(j)})^T S'' D_{W_{:,j}} A_{:,j}$$

so $\tilde{U}$ and $\tilde{V}$ can be encoded as rational functions over $\Theta(\frac{(s+\log(1/\varepsilon))kr}{\varepsilon})$ variables by Cramer's Rule.

By Theorem 2, we can argue that $\min_{\tilde{U}, \tilde{V}} \|W \circ (\tilde{U}\tilde{V} - A)\|_F^2 + \lambda \|\tilde{U}\|_F^2 + \lambda \|\tilde{V}\|_F^2$ is a good approximation for $\|W \circ (U^* V^* - A)\|_F^2 + \lambda \|U^*\|_F^2 + \lambda \|V^*\|_F^2$ with constant probability, and so in particular such a good approximation exists. By using the polynomial system feasibility checkers described in Theorem 1 and following similar procedures and doing binary search, we get an polynomial system with $O(nk)$-degree and $O(\frac{s+\log(1/\varepsilon)}{\varepsilon}kr)$ variables after simplifying and so our polynomial solver runs in time $n^{O((s+\log(1/\varepsilon))kr/\varepsilon)} \log^{O(1)}(\Delta/\delta)$.

**Theorem 3.** *Given matrices $A, W \in \mathbb{R}^{n \times d}$ and $\varepsilon < 0.1$ such that*

1. *rank(W) = r*

2. *non-zero entries of $A, W$ are multiples of $\delta > 0$*

3. *all entries of $A, W$ are at most $\Delta$ in absolute value*

4. $s = \max_{i,j}\{sd_\lambda(V^* D_{W_{i,:}}), sd_\lambda(D_{W_{:,j}} U^*)\} < k$

*there is an algorithm to find $U \in \mathbb{R}^{n \times k}, V \in \mathbb{R}^{k \times d}$ in time $n^{O((s+\log(1/\varepsilon))kr/\varepsilon)} \log^{O(1)}(\Delta/\delta)$ such that $\|W \circ (UV - A)\|_F^2 + \lambda \|U\|_F^2 + \lambda \|V\|_F^2 \le (1+\varepsilon)\text{OPT}$.*

## 4.2 Removing Rank Dependence

Note that the running time of our algorithm still depends on $k$, the dimension that we are reducing to. To remove this, we prove a structural theorem about low rank approximation of low statistical dimension matrices.

**Theorem 4.** *Given matrices $A, W$ in $\mathbb{R}^{n \times d}$ and $\varepsilon < 0.1$ such that rank(W) is $r$, and letting $s$ equal $\max_{i,j}\{sd_\lambda(V^* D_{W_{i,:}}), sd_\lambda(D_{W_{:,j}} U^*)\} < k$, if we let $\text{OPT}(k)$ denote*

$$\min_{U \in \mathbb{R}^{n \times k}, V \in \mathbb{R}^{k \times d}} \|W \circ (UV - A)\|_F^2 + \lambda \|U\|_F^2 + \lambda \|V\|_F^2$$

*then $\text{OPT}(O(r(s + \log(1/\varepsilon))/\varepsilon)) \le (1+\varepsilon)\text{OPT}(k)$*

Combining Theorem 3 and Theorem 4, we have our final theorem. We note that this also improves running time bounds of un-weighted regularized low rank approximation in Section 3 of [ACW17].

**Theorem 5.** *Given matrices $A, W \in \mathbb{R}^{n \times d}$ and $\varepsilon < 0.1$ and the conditions of Theorem 3, there is an algorithm to find $U \in \mathbb{R}^{n \times k}, V \in \mathbb{R}^{k \times d}$ in time $n^{O(r^2(s+\log(1/\varepsilon))^2/\varepsilon^2)} \log^{O(1)}(\Delta/\delta)$ such that $\|W \circ (UV - A)\|_F^2 + \lambda \|U\|_F^2 + \lambda \|V\|_F^2 \leq (1+\varepsilon)\mathrm{OPT}$.*

# 5 Reducing the Degree of the Solver

## 5.1 Non-negative Weight Matrix and Non-Negative Rank

Under the case where $W$ is rank $r$ with only $r$ distinct columns (up to scaling), we are able to improve the running time to $poly(n)2^{r^3(s+\log(1/\varepsilon))^2/\varepsilon^2}$ by showing that the degree of the solver is $O(rk)$ as opposed to $O(nk)$. Specifically, the $O(nk)$ degree comes from clearing the denominator of the rational expressions that come from naïvely using and analyzing Cramer's Rule; in this section, we demonstrate different techniques to avoid the dependence on $n$. We also show the same running time bound under a more relaxed assumption of non-negative rank, which is always less than or equal to the number of distinct columns.

**Theorem 6.** *Given matrices $A, W \in \mathbb{R}^{n \times d}$ and $\varepsilon < 0.1$ and suppose the conditions of Theorem 3 hold. Furthermore, we are given $Y, Z \geq 0$ such that $W = YZ$ and $Y, Z^T$ has $\mathtt{nnr}(W) = r'$ columns.*

*Then, there is an algorithm to find $U \in \mathbb{R}^{n \times k}, V \in \mathbb{R}^{k \times d}$ in time $poly(n) \cdot 2^{O(r'r^2(s+\log(\frac{1}{\varepsilon}))^2 \frac{1}{\varepsilon^2})} \cdot \log^{O(1)}\left(\frac{\Delta}{\delta}\right)$ such that $\|W \circ (UV - A)\|_F^2 + \lambda \|U\|_F^2 + \lambda \|V\|_F^2 \leq (1+\varepsilon)\mathrm{OPT}$.*

## 5.2 Richardson's Iteration

Note that the current polynomial solver uses Cramer's rule to solve

$$\tilde{U} = \underset{U \in \mathbb{R}^{n \times k}}{\arg\min} \sum_{i=1}^{n} \|U_{i,:} P^{(i)} - A_{i,:} D_{W_{i,:}} S'\|^2 + \lambda \|U_{i,:}\|^2$$

giving

$$\tilde{U}_{i,:} = A_{i,:} D_{W_{i,:}} S' (P^{(i)})^T (P^{(i)}(P^{(i)})^T + \lambda I_k)^{-1}.$$

We want to use Richardson's iteration instead to avoid rational expressions and the dependence on $n$ in the degree that comes from clearing the denominator.

**Theorem 7** (Preconditioned Richardson [CKP$^+$17])**.** *Let $A, B$ be symmetric PSD matrices such that $ker(A) = ker(B)$ and $\eta A \preceq B \preceq A$. Then, for any $b$, if $x_0 = 0$ and $x_{i+1} = x_i - \eta B^{-1}(Ax_i - b)$,*

$$\|x_t - A^{-1}b\| \leq \varepsilon \|A^{-1}b\|$$

*for $t = \Omega(\log(c_B/\varepsilon)/\eta)$. Furthermore, we may express $x_t$ as a polynomial of degree $O(t)$ in terms of the entries of $B^{-1}$ and $A$.*

**Theorem 8.** *Given matrices $A, W \in \mathbb{R}^{n \times d}$ and $\varepsilon < 0.1$ and suppose the conditions of Theorem 3 hold. Furthermore, let $\sigma = \max_{i,j}\{\sigma_1(V^* D_{W_{i,:}}), \sigma_1(D_{W_{:,j}} U^*)\}$.*

*There is an algorithm to find $U \in \mathbb{R}^{n \times k}, V \in \mathbb{R}^{k \times d}$ in time $poly(n)\left(\frac{\sigma^2}{\lambda} \cdot \log\left(\frac{\Delta(\sigma^2+\lambda)n}{\lambda\tau}\right)\right)^l \cdot \log^{O(1)}\left(\frac{\Delta}{\delta}\right)$, where $l = O((s+\log(\frac{1}{\varepsilon}))^2 \frac{r^2}{\varepsilon^2})$, such that $\|W \circ (UV - A)\|_F^2 + \lambda \|U\|_F^2 + \lambda \|V\|_F^2 \leq (1+\varepsilon)\mathrm{OPT} + \tau$.*

## 5.3 Preconditioned GD

Instead of directly using Richardson's iteration, we may use a preconditioner first instead. The right preconditioner can also be guessed at a cost of increasing the number of variables. Note that multiple preconditioners may be used, but for now, we demonstrate the power of a single preconditioner.

**Theorem 9.** *Given matrices $A, W \in \mathbb{R}^{n \times d}$ and $\varepsilon < 0.1$ and suppose the conditions of Theorem 8 hold. Furthermore, $0 < l_W \leq |W| \leq u_W$. Then, there is an algorithm to find $U \in \mathbb{R}^{n \times k}, V \in \mathbb{R}^{k \times d}$ in time $poly(n) \cdot \left(\frac{u_W}{l_W} \cdot \log\left(\frac{\Delta(\sigma^2+\lambda)n}{\lambda\tau}\right)\right)^l \cdot \log^{O(1)}\left(\frac{\Delta}{\delta}\right)$, where $l = O((s+\log(\frac{1}{\varepsilon}))^2 \frac{r^2}{\varepsilon^2})$, such that $\|W \circ (UV - A)\|_F^2 + \lambda \|U\|_F^2 + \lambda \|V\|_F^2 \leq (1+\varepsilon)\mathrm{OPT} + \tau$.*

# 6  Experiments

The goal of our experiments was to show that sketching down to the statistical dimension can be applied to regularized weighted low rank approximation without sacrificing overall accuracy in the objective function, as our theory predicts. We combine sketching with a common practical alternating minimization heuristic for solving regularized weighted low rank approximation, rather than implementing a polynomial system solver. At each step in the algorithm, we have a candidate $U$ and $V$ and we perform a "best-response" where we either update $U$ to give the best regularized weighted low rank approximation cost for $V$ or we update $V$ to give the best regularized weighted low rank approximation cost for $U$. We used a synthetic dataset and several real datasets (connectus, NIPS, landmark, and language) [DH11, PJST17]. All our experiments ran on a MacBook Pro 2012 with 8GB RAM and a 2.5GHz Intel Core i5 processor.

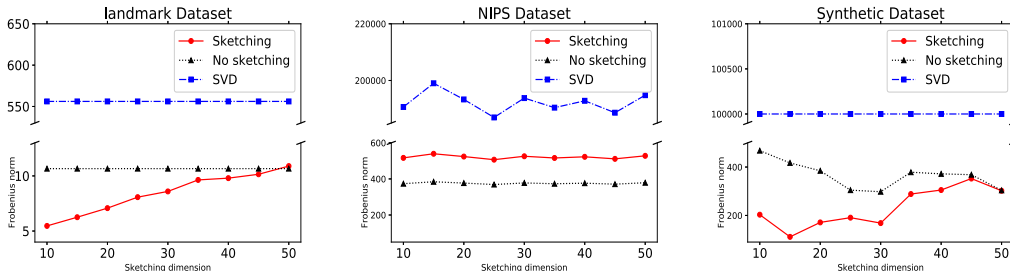

Figure 1: Regularized weighted low-rank approximations with $\lambda = 0.556$ for landmark, $\lambda = 314$ for NIPS, and $\lambda = 1$ for the synthetic dataset.

For all datasets, the task was to find a rank $k = 50$ decomposition of a given matrix $A$. For the experiments of Figure 1 and Figure 2, we generated dense weight matrices $W$ with the same shape as $A$ and with each entry being a 1 with probability 0.8, a 0.1 with probability 0.15, and a 0.01 with probability 0.05. For the experiments of Figure 3, we generated binary weight matrices where each entry was 1 with probability 0.9. Note that this setting corresponds to a regularized form of matrix completion. We set the regularization parameter $\lambda$ to a variety of values (described in the Figure captions) to illustrate the performance of the algorithm in different settings.

For the synthetic dataset, we generated matrices $A$ with dimensions $10000 \times 1000$ by picking random orthogonal vectors as its singular vectors and having one singular value equal to 10000 and making the rest small enough so that the statistical dimension of $A$ would be approximately 2.

For the real datasets, we chose the connectus, landmark, and language datasets [DH11] and the NIPS dataset [PJST17]. We sampled 1000 rows from each adjacency or word count matrix to form a matrix $B$ and then let $A$ be the radial basis function kernel of $B$. We performed three algorithms on each dataset: Singular Value Decomposition, Alternating Minimization without Sketching, and Alternating Minimization with Sketching. We parameterized the experiments by $t$, the sketch size, which took values in $\{10, 15, 20, 25, 30, 35, 40, 45, 50\}$. For each value of $t$ we generated a weight matrix and either generated a synthetic dataset or sampled a real dataset as described in the above paragraphs, then tested our three algorithms.

For the SVD, we just took the best rank $k$ approximation to $A$ as given by the top $k$ singular vectors. We used the built-in `svd` function in numpy's linear algebra package.

For Alternating Minimization without Sketching, we initialized the low rank matrix factors $U$ and $V$ to be random subsets of the rows and columns of $A$ respectively, then performed $n = 25$ steps of alternating minimization.

For Alternating Minimization with Sketching, we initialized $U$ and $V$ the same way, but performed $n = 25$ best response updates in the sketched space, as in Theorem 3. The sketch $S$ was chosen to be a CountSketch matrix with $t$. Based on Theorem 5, we calculated a rank $t < k$ approximation of $A$ whenever we used a sketch of size $t$. We plotted the objective value of the low rank approximation for the connectus, NIPS, and synthetic datasets (the other datasets as well as a different family of weight matrices are discussed in the supplementary material) for each value of $t$ and each algorithm in Figure 1. The experiment with the landmark dataset in Figure 1 used a regularization parameter

value of $\lambda = 0.556$, while the experiments with the NIPS and synthetic datasets used a value of $\lambda = 1$. Objective values were given in 1000's in the Frobenius norm.

Both forms of alternating minimization greatly outperform the low rank approximation given by the SVD. Alternating minimization with sketching comes within a factor of 1.5 approximation to alternating minimization without sketching and can sometimes slightly outperform alternating minimization without sketching[2], showing that performing CountSketch at each best response step does not result in a critically suboptimal objective value. The runtime of alternating minimization with sketching varies from being around 2 times as fast as alternating minimization without sketching (when the sketch size $t = 10$) to being around 1.4 times as fast (when the sketch size $t = 50$). Table 1 shows the runtimes for the non-synthetic experiments of Figure 1.

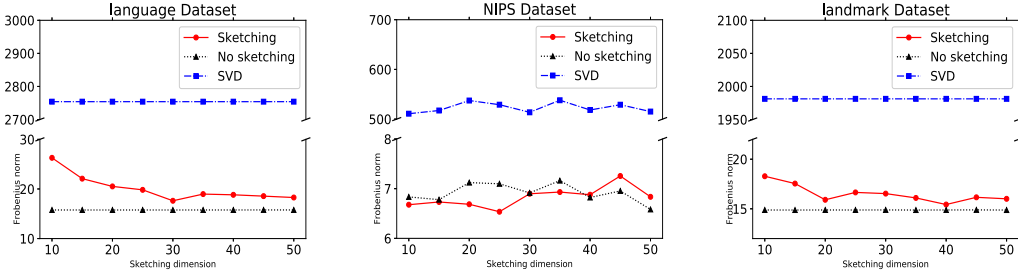

Figure 2: Regularized weighted low-rank approximations with $\lambda = 2.754$ for language, $\lambda = 1$ for NIPS, and $\lambda = 1.982$ for landmark.

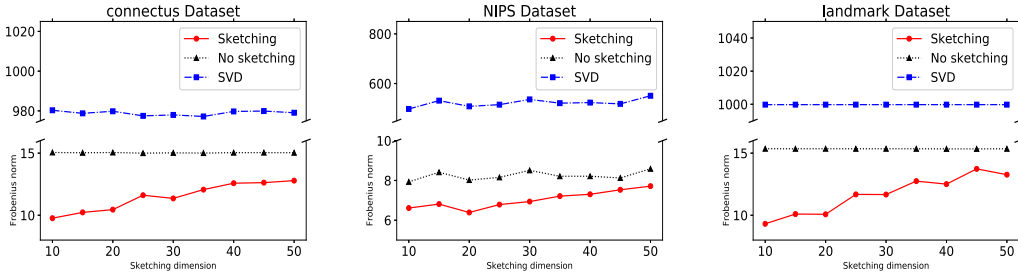

Figure 3: Regularized weighted low-rank approximations with binary weights and $\lambda = 1$.

| Runtimes w/ sketching | | | Runtimes wo/ sketching | | |
|---|---|---|---|---|---|
| t | landmark | NIPS | t | landmark | NIPS |
| 10 | 54.31 | 49.1 | 10 | 126.22 | 104.5 |
| 15 | 53.58 | 50.33 | 15 | 113.8 | 105.75 |
| 20 | 57.65 | 51.8 | 20 | 119.17 | 104.28 |
| 25 | 65.53 | 56.43 | 25 | 121.69 | 104.35 |
| 30 | 68.68 | 57.34 | 30 | 123.51 | 105.42 |
| 35 | 72.22 | 62.66 | 35 | 129.87 | 100.5 |
| 40 | 79.94 | 63.48 | 40 | 123.65 | 101.75 |
| 45 | 81.22 | 67.73 | 45 | 109.02 | 104.93 |
| 50 | 72.77 | 73.11 | 50 | 100.61 | 101.77 |

Table 1: Runtimes in seconds for alternating minimization with and without sketching.

**Acknowledgements:** Part of this work was done while D. Woodruff was visiting Google Mountain View as well as the Simons Institute for the Theory of Computing, and was supported in part by an Office of Naval Research (ONR) grant N00014-18-1-2562.

## Footnotes

[1]https://en.wikipedia.org/wiki/Low-rank_approximation#Weighted_low-rank_approximation_problems

[2]See the supplementary material for additional discussion.

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
