[Supplementary Material · appendix 2243.pdf]

# A Proof of Theorem 2

*Proof.* Let $\widehat{S} = \begin{bmatrix} S & 0 \\ 0 & I_k \end{bmatrix} \in \mathbb{R}^{(\ell+k)\times(n+k)}$, $\widehat{M}^{(i)} = \begin{bmatrix} M^{(i)} \\ \sqrt{\lambda}I_k \end{bmatrix} \in \mathbb{R}^{(n+k)\times k}$, and $\widehat{b}^{(i)} = \begin{bmatrix} b^{(i)} \\ 0 \end{bmatrix} \in \mathbb{R}^{n+k}$. Observe that $\|M^{(i)}x - b^{(i)}\|^2 + \lambda\|x\|^2 = \|\widehat{M}^{(i)}x - \widehat{b}^{(i)}\|^2$ and $\|S(M^{(i)}y - b^{(i)})\|^2 + \lambda\|y\|^2 = \|\widehat{S}(\widehat{M}^{(i)}y - \widehat{b}^{(i)})\|^2$.

It suffices to prove that, for $1 \leq i \leq d$,

$$\mathbf{E}_S\left[\|\widehat{M}^{(i)}y^{(i)} - \widehat{b}^{(i)}\|^2 - \|\widehat{M}^{(i)}x^{(i)} - \widehat{b}^{(i)}\|^2\right] \leq O(\varepsilon) \cdot \|\widehat{M}^{(i)}x^{(i)} - \widehat{b}^{(i)}\|^2$$

because we can sum over all $i$ and apply Markov's inequality to complete the argument.

Fix some $i$ and set $M = M^{(i)}, b = b^{(i)}, \widehat{M} = \widehat{M}^{(i)}, \widehat{b} = \widehat{b}^{(i)}, x = x^{(i)}, y = y^{(i)}$. Let $\widehat{U}_b$ be an orthogonal matrix whose columns form an orthonormal basis for the columns of $[\widehat{M}\ \widehat{b}]$. Now define $\Gamma := \|\widehat{U}_b^T \widehat{S}^T \widehat{S} \widehat{U}_b - I_{k+1}\|$. We let $U_1$ form its first $n$ rows and $U_2$ form the rest.

Since $s < k$, we can have an unbounded condition number if we just look at $c_{\widehat{S}}(\widehat{M})$ so we need to have a more subtle analysis than in [RSW16]. Instead of simply conditioning on $\Gamma$, let us define $M = M_h + M_t$, where $M_h$ is the component of $M$ corresponding to the span of singular vectors with values that are $\sigma_i^2 \geq \lambda$. Then, $M_t$ is the orthogonal component to $M_h$ and is a subspace of the span of singular vectors corresponding to values that are $\sigma_i^2 < \lambda$. Since $2 \geq 1 + \frac{\lambda}{\sigma_i^2}$, for $\sigma_i$ corresponding to $M_h$, then the rank of $M_h$ is bounded by $r_h = O(\mathtt{sd}_\lambda(M))$. Since $S$ has at least $\Omega(\mathtt{sd}_\lambda(M)/\varepsilon)$ rows, with probability 1, the condition number $c_h = c_S([M_h, b])$ will be finite with probability 1 (note that $b$ can be assumed to be orthogonal to the image of $M$).

Let $\alpha = c_h(1 + \Gamma)$ be a product of condition numbers. If $\alpha$ is close to 1, then we are in a good regime so we will condition on $\alpha$. It follows that

$$\mathbf{E}_S\left[\|\widehat{M}y - \widehat{b}\|^2 - \|\widehat{M}x - \widehat{b}\|^2\right] = \mathbf{Pr}[\alpha > 1.1] \cdot \mathbf{E}_S\left[\|\widehat{M}y - \widehat{b}\|^2 - \|\widehat{M}x - \widehat{b}\|^2 \big| \alpha > 1.1\right]$$
$$+ \mathbf{Pr}[\alpha \leq 1.1] \cdot \mathbf{E}_S\left[\|\widehat{M}y - \widehat{b}\|^2 - \|\widehat{M}x - \widehat{b}\|^2 \big| \alpha \leq 1.1\right]$$

We will now bound the two terms in the sum and our final result follows by combining Claim A.1 and Claim A.4.

**Claim A.1.**

$$\mathbf{Pr}[\alpha > 1.1]\,\mathbf{E}_S\left[\|\widehat{M}y - \widehat{b}\|^2 - \|\widehat{M}x - \widehat{b}\|^2 \big| \alpha > 1.1\right] \leq O(\varepsilon) \cdot \|\widehat{M}x - \widehat{b}\|^2$$

*Proof.* To bound our expression, note that

$$\|\widehat{M}y - \widehat{b}\|^2 \leq \frac{1}{\kappa_{\widehat{S}}([\widehat{M}\ \widehat{b}])}\|\widehat{S}(\widehat{M}y - \widehat{b})\|^2 \leq \frac{1}{\kappa_{\widehat{S}}([\widehat{M}\ \widehat{b}])}\|\widehat{S}(\widehat{M}x - \widehat{b})\|^2 \leq \frac{K_{\widehat{S}}([\widehat{M}\ \widehat{b}])}{\kappa_{\widehat{S}}([\widehat{M}\ \widehat{b}])}\|\widehat{M}x - \widehat{b}\|^2$$

By Claim A.2, $\|\widehat{M}y - \widehat{b}\|^2 \leq 10\alpha^2\|\widehat{M}x - \widehat{b}\|^2$. By Claim A.3,

$$\mathbf{E}_S\left[\|\widehat{M}y - \widehat{b}\|^2 - \|\widehat{M}x - \widehat{b}\|^2 \big| \alpha > 1.1\right] \leq 10\|\widehat{M}x - \widehat{b}\|^2 \,\mathbf{E}_S\left[\alpha^2 \big| \alpha > 1.1\right] \leq O(\varepsilon)\|\widehat{M}x - \widehat{b}\|^2$$

$\square$

**Claim A.2.** $K_{\widehat{S}}([\widehat{M}\ \widehat{b}]) \leq \alpha$ and $\frac{1}{\kappa_{\widehat{S}}([\widehat{M}\ \widehat{b}])} \leq 10\alpha$

*Proof.* First, we bound $K_{\widehat{S}}([\widehat{M}\ \widehat{b}])$. By definition of $\Gamma$, $\|S(Mx - b)\|^2 + \lambda\|x\|^2 \leq (1+\Gamma)(\|Mx - b\|^2 + \lambda\|x\|^2)$ so $K_{\widehat{S}}([\widehat{M}\ \widehat{b}]) \leq 1 + \Gamma \leq \alpha$.

More importantly, we want to bound $\frac{1}{\kappa_{\widehat{S}}([\widehat{M}\ \widehat{b}])}$. First, we claim that $\|SM_tx\| \leq \sqrt{\lambda(1+2\Gamma)}\|x\|$. We may assume $x$ lies entirely in the column space of $M_t$ and by definition of $M_t$, we know that

$\|Mx\|^2 = \|M_t x\|^2 \le \lambda\|x\|^2$. Now, by the definition of $\Gamma$, $\|SM_t x\|^2 = \|SMx\|^2$ which is at most $(1+\Gamma)\|Mx\|^2 + \Gamma\lambda\|x\|^2 \le (1+2\Gamma)\lambda\|x\|^2$.

We now consider two cases: one where $\|S(M_h x - b)\|$ is at least $2\sqrt{\lambda(1+2\Gamma)}\|x\|$ and one where $\|S(M_h x - b)\| < 2\sqrt{\lambda(1+2\Gamma)}\|x\|$.

For all $x$ such that $\|S(M_h x - b)\| \ge 2\sqrt{\lambda(1+2\Gamma)}\|x\|$, we rewrite $\|S(Mx - b)\|^2 = \|S(M_h x - b) + SM_t x\|^2$. Then, by Cauchy-Schwarz,

$$\|S(Mx - b)\|^2 \ge \|S(M_h x - b)\|^2 - 2|\langle S(M_h x - b), SM_t x\rangle|$$
$$\ge \|S(M_h x - b)\|^2 - 2\|S(M_h x - b)\|\|SM_t x\| = \|S(M_h x - b)\|(\|S(M_h x - b)\| - \|SM_t x\|)$$
$$\ge 0.5\|S(M_h x - b)\|^2 \ge (0.5/c_h)\|M_h x - b\|^2,$$

where the fourth line follows since $\|S(M_h x - b)\| \ge 2\sqrt{\lambda(1+2\Gamma)}\|x\| \ge 2\|SM_t x\|$ and the fifth line follows from definition of $c_h$.

Finally, this implies

$$\|S(Mx - b)\|^2 + \lambda\|x\|^2 \ge (0.5/c_h)(\|M_h x - b\|^2 + 2\lambda\|x\|^2)$$
$$\ge (0.5/c_h)(\|M_h x - b\|^2 + \|M_t x\|^2 + \lambda\|x\|^2) \ge (1/2c_h)(\|Mx - b\|^2 + \lambda\|x\|^2)$$

where the first line follows since $c_h > 1$, the second line follows from $\|M_t x\|^2 < \lambda\|x\|^2$ and the last line from orthogonality.

Now consider all $x$ such that $\|S(M_h x - b)\|$ is less than $2\sqrt{\lambda K_S(M)}\|x\|$. This means $\|M_h x - b\|$ is less than $2\sqrt{c_h \lambda K_S(M)}\|x\|$. Then,

$$\|Mx - b\|^2 + \lambda\|x\|^2 \le 4c_h(1+2\Gamma)\lambda\|x\|^2 + \lambda\|x\|^2$$
$$\le \frac{1}{4c_h(1+2\Gamma)+1} \cdot \lambda\|x\|^2 \le \frac{1}{5c_h(1+2\Gamma)} \cdot (\|S(Mx - b)\|^2 + \lambda\|x\|^2)$$

Together, we conclude that $\frac{1}{\kappa_{\widehat{S}}([\widehat{M}\,\widehat{b}])} \le \max(5c_h(1+2\Gamma), 2c_h) \le 10\alpha$, where $\alpha = c_h(1+\Gamma)$.

$\square$

**Claim A.3.**
$$\mathbf{Pr}\left[\alpha > 1.1\right]\underset{S}{\mathbf{E}}\left[\alpha^2\Big|\alpha > 1.1\right] = O(\varepsilon)$$

*Proof.* Note that for $t > 1$, we have

$$\mathbf{Pr}\left(\alpha > t\right) \le \mathbf{Pr}\left(1 + \Gamma > \sqrt{t}\right) + \mathbf{Pr}\left(1 + \Gamma \le \sqrt{t} \text{ and } c_h > \sqrt{t}\right)$$
$$\le \mathbf{Pr}\left(1 + \Gamma > \sqrt{t}\right) + \mathbf{Pr}\left(c_h > \sqrt{t}\right)$$

By Lemma 12 of [ACW17], note that $\|U_1\|_F^2$ is at most $\mathtt{sd}_\lambda(M) + 1$ and $\|U_1\| < 1$. Now, we can express $\Gamma = \|\widehat{U}_b^T \widehat{S}^T \widehat{S}\widehat{U}_b - I_{k+1}\|$ which is equal to $\|U_1^T S^T S U_1 - U_1^T U_1\|$. By Lemma 2.1 with $A = B = U_1$ and $\gamma = \frac{\sqrt{t}}{\|U_1\|^2}$, then for any $t > 1.1$, we have

$$\mathbf{Pr}[1 + \Gamma > \sqrt{t}] < \varepsilon t^{-\Omega(1)}$$

since $\ell$ is larger than $\Omega(\frac{1}{\gamma^2}(\|U_1\|_F^2/\|U_1\|^2 + \log(t/\varepsilon))) = \Omega(\mathtt{sd}_\lambda(M) + \log(1/\varepsilon))$.

Then, by [CD08], since $M_h$ only has less than $O(\mathtt{sd}_\lambda(M))$ columns, then since $\ell > \mathtt{sd}_\lambda(M)/\varepsilon$, we have for $t > 1.1$, $\mathbf{Pr}[c_S(M) > \sqrt{t}] = \Theta(t^{-1/\varepsilon}) < \varepsilon t^{-\Omega(1)}$. Thus, we conclude that

$$\mathbf{Pr}\left[\alpha > 1.1\right]\underset{S}{\mathbf{E}}\left[\alpha^2\Big|\alpha > 1.1\right] \le O(1)\mathbf{Pr}[\alpha > 1.1] + \int_{1.1}^{\infty} t\,\mathbf{Pr}[\alpha > t]\,dt = O(\varepsilon)$$

$\square$

**Claim A.4.** $\mathbf{E}_S\left[\|\widehat{M}y - \widehat{b}\|^2 - \|\widehat{M}x - \widehat{b}\|^2\Big|\alpha \le 1.1\right] \le O(\varepsilon) \cdot \|\widehat{M}x - \widehat{b}\|^2$

*Proof.* The normal equations for $x$ tell us that $\widehat{M}^T(\widehat{M}x - \widehat{b})$ is 0. Thus, by the Pythagorean Theorem,
$$\|\widehat{M}y - \widehat{b}\|^2 - \|\widehat{M}x - \widehat{b}\|^2 = \|\widehat{M}(y-x)\|^2 = \|\tilde{y} - \tilde{x}\|^2$$
where $\widehat{U}\tilde{y} = \widehat{M}y$ and $\widehat{U}\tilde{x} = \widehat{M}x$.

We have $\|\tilde{y} - \tilde{x}\| \leq \|(\widehat{U}^T\widehat{S}^T\widehat{S}\widehat{U} - I_k)(\tilde{y} - \tilde{x})\| + \|\widehat{U}^T\widehat{S}^T\widehat{S}\widehat{U}(\tilde{y} - \tilde{x})\|$, so since we are conditioning on $\alpha \leq 1.1$, we know that $\Gamma \leq 0.1$, which implies that $\|\tilde{y} - \tilde{x}\| \leq O(1)\|\widehat{U}^T\widehat{S}^T\widehat{S}\widehat{U}(\tilde{y} - \tilde{x})\|$.

Since $\mathbf{Pr}[\alpha \leq 1.1] \geq 1 - O(\varepsilon)$, then
$$\mathop{\mathbf{E}}_{S}\left[\|\widehat{M}y - \widehat{b}\|^2 - \|\widehat{M}x - \widehat{b}\|^2 \big| \alpha \leq 1.1\right] \leq O(1) \cdot \mathop{\mathbf{E}}_{S}\left[\|\widehat{U}^T\widehat{S}^T\widehat{S}\widehat{U}(\tilde{y} - \tilde{x})\|^2\right].$$

and the normal equations for $\tilde{y}$ tell us that $\widehat{U}^T\widehat{S}^T\widehat{S}(\widehat{U}\tilde{y} - \widehat{b})$ is 0. Thus,
$$\mathop{\mathbf{E}}_{S}\left[\|\widehat{U}^T\widehat{S}^T\widehat{S}\widehat{U}(\tilde{y} - \tilde{x})\|^2\right] = \mathop{\mathbf{E}}_{S}\left[\|\widehat{U}^T\widehat{S}^T\widehat{S}(\widehat{U}\tilde{x} - \widehat{b})\|^2\right].$$

Let $t$ be a natural number. Note that $S$ has $\Omega(\frac{1}{\varepsilon}(s + \log(1/\varepsilon)) = \Omega(\frac{1}{t\varepsilon}(s + \log((1/\varepsilon)^t))$ rows. Note that $\widehat{S}$ only sketches $U_1$ but leaves $U_2$ un-sketched. By Lemma 2.1 with $A = U_1$, $B = U_1\tilde{x} - b$, $\gamma = \sqrt{t\varepsilon}/\|U_1\|$ we have
$$\mathbf{Pr}\left[\|\widehat{U}^T\widehat{S}^T\widehat{S}(\widehat{U}\tilde{x} - \widehat{b})\| > \sqrt{t\varepsilon}\|\widehat{U}\tilde{x} - \widehat{b}\|\right] < O(\varepsilon^t) \tag{2}$$

Let $E_t$ denote the event that $\|\widehat{U}^T\widehat{S}^T\widehat{S}(\widehat{U}\tilde{x} - \widehat{b})\|$ is between $\sqrt{(t-1)\varepsilon}\|\widehat{M}x - \widehat{b}\|$ and $\sqrt{t\varepsilon}\|\widehat{M}x - \widehat{b}\|$. By inequality (2) we have
$$\mathop{\mathbf{E}}_{S}\left[\|\widehat{U}^T\widehat{S}^T\widehat{S}(\widehat{U}\tilde{x} - \widehat{b})\|^2\right] \leq \|\widehat{M}x - \widehat{b}\|^2 \sum_{t=1}^{\infty} t\varepsilon \cdot \mathbf{Pr}[E_t\|]$$

$$\leq \varepsilon \cdot \|\widehat{M}x - \widehat{b}\|^2 \sum_{t=1}^{\infty} O(\varepsilon^t) \cdot t \leq O(\varepsilon) \cdot \|\widehat{M}x - \widehat{b}\|^2$$

and we are done. $\qquad\square$

$\square$

# B  Proof of Theorem 4

*Proof.* Observe that our algorithm in Theorem 3,
$$\tilde{V} = \operatorname*{argmin}_{V \in \mathbb{R}^{k \times n}} \sum_{j=1}^{d} \|Q^{(j)}V_{:,j} - S''D_{W_{:,j}}A_{:,j}\|^2 + \lambda\|V_{:,j}\|^2$$

where $Q^{(j)} = S''D_{W_{:,j}}U$ which equals $R^{(j)}U$ which is a $O((s + \log(1/\varepsilon))/\varepsilon)$ by $k$ matrix and $R^{(j)} = S''D_{W_{:,j}}$. Note that $R^{(j)}$ can be written as a linear combination of $r$ matrices with rank at most $O((s + \log(1/\varepsilon))/\varepsilon)$. Therefore, by letting $P$ be the projection matrix on the span of the total
$$O((s + \log(1/\varepsilon))r/\varepsilon)$$
right singular vectors of these $r$ matrices, we see that $\tilde{V}$ equals
$$\operatorname*{argmin}_{V \in \mathbb{R}^{k \times n}} \sum_{j=1}^{d} \|R^{(j)}PUV_{:,j} - S''D_{W_{:,j}}A_{:,j}\|^2 + \lambda\|V_{:,j}\|^2$$

Since this holds for any $U$, we see that
$$\|W \circ (PU^*\tilde{V} - A)\|_F^2 + \lambda\|PU^*\|_F^2 + \lambda\|\tilde{V}\|_F^2$$
$$\leq \|W \circ (U^*\tilde{V} - A)\|_F^2 + \lambda\|U^*\|_F^2 + \lambda\|\tilde{V}\|_F^2$$

Since $PU^*\tilde{V}$ has rank at most the rank of $P$, we conclude by noting that by the guarantees of Theorem 3,
$$\|W \circ (U^*\tilde{V} - A)\|_F^2 + \lambda\|U^*\|_F^2 + \lambda\|\tilde{V}\|_F^2 \leq (1 + \varepsilon)\mathrm{OPT}(k)$$

$\square$

# C  Proof of Theorem 6

*Proof.* Since we know $W = YZ$, where $Y \in R^{n \times r'}$ and $Z \in \mathbb{R}^{r' \times n}$ are non-negative, then we claim that we have a rounding procedure to create $W'$ with only $(\log n/\varepsilon)^{r'}$ distinct columns and rows that produces an $\varepsilon$-close solution. The procedure is as expected: round all values in $Y, Z$ to the nearest power of $(1+\varepsilon)$ and call $W' = Y'Z'$ our new matrix. Note that there are at most $(\log n/\varepsilon)^{r'}$ distinct rows, since each row in $A$ takes on only $(\log n/\varepsilon)^{r'}$ possible values. Symmetrically, the number of columns is bounded by the same value. Now, we claim that:

**Claim C.1.** $(1 - \varepsilon)^2 W \leq W' \leq (1 + \varepsilon)^2 W$

*Proof.* It suffices to show that the intermediary matrix $\widehat{W} = Y'Z$ satisfies this bound. Consider each row of $\widehat{W}$, so WLOG, let $\widehat{W}_1$ be the first row of $\widehat{W}$ which can be expressed as $\widehat{W}_1 = \sum_{i=1}^{r'} (Y')_{1i} Z_i$, where $Z_i$ is the i-th row of $Z$. Note that $W_1 = \sum_{i=1}^{r'} Y_{1i} Z_i$. Finally, since $(Y')_{1i} \in (1-\varepsilon, 1+\varepsilon) Y_{1i}$ by our rounding procedure and all values in $Y_{1i}, Z_i$ are non-negative, we deduce that $(1-\varepsilon)\widehat{W}_1 \leq W_1 \leq (1+\varepsilon)\widehat{W}_1$. $\qquad\square$

Since we only have $(\log n/\varepsilon)^{r'}$ distinct columns and rows, when we call the polynomial solver, the degree of our system is at most $(\log n/\varepsilon)^{r'}$ and our bound follows from polynomial system solver guarantees. $\qquad\square$

# D  Proof of Theorem 8

*Proof.* Fix some $i$. By Theorem 7, we may express a $k$-order approximation of $\tilde{U}_{i,:}$ as

$$\tilde{U}_{i,:}^k = A_{i,:} D_{W_{i,:}} S'(P^{(i)})^T p_k(P^{(i)}(P^{(i)})^T + \lambda I_k)$$

where $p_k$ is a degree $O(k)$ polynomial that approximate the inverse. Furthermore, we claim $\lambda I_k \preceq P^{(i)}(P^{(i)})^T + \lambda I_k \preceq \log(n)(1 + \sigma/\lambda)I_k$, where $P^{(i)} = V^* D_{w_i,:} S$, with high probability.

By the same arguments as in the proof of Theorem 2 in Claim A.3, let $M^T = V^* D_{w_i,:}$ and $\widehat{M}$ defined analogously, along with $\widehat{U}, \widehat{S}$. Now define $\Gamma := \|\widehat{U}^T \widehat{S}^T \widehat{S}\widehat{U} - I_k\|$. Again, let $U_1$ form its first $n$ rows and $U_2$ form the rest. Note that $\|U_1\|_F^2 \leq \text{sd}_\lambda(M)$ and $\|U_1\| < 1$. By Lemma 2.1 with $A = B = U_1$ and $\gamma = \log(n)/\|U_1\|^2$, then we have

$$\mathbf{Pr}[1 + \Gamma > \log(n)] < n^{-\Omega(1)}$$

since $\ell > \Omega(\frac{1}{\gamma^2}(\|U_1\|_F^2/\|U_1\|^2 + \log(n))) = \Omega(\text{sd}_\lambda(M)) = \Omega(s)$.

This implies that with high probability

$$\|SMx\|^2 + \lambda\|x\|^2 \leq \log(n)(\|Mx\|^2 + \lambda\|x\|^2).$$

Specifically, we have $\sigma_1(P^{(i)})^2 \leq \log(n)\sigma_1(V^* D_{w_i,:})^2 + \lambda \log(n) \leq \log(n)\sigma^2 + \lambda \log(n)$.

Therefore, by the guarantees of Theorem 7, we have

$$\tilde{U}_{i,:}^k P^{(i)} - A_{i,:} D_{W_{i,:}} S'\|^2 + \lambda\|\tilde{U}_{i,:}^k\|^2 \leq \|\tilde{U}_{i,:} P^{(i)} - A_{i,:} D_{W_{i,:}} S'\|^2 + \lambda\|\tilde{U}_{i,:}\|^2 + \tau/n$$

holds if $k > \Omega((\sigma^2/\lambda)\log(\Delta(\sigma^2 + \lambda)/\lambda n/\tau))$.

Summing up for all $i$ and applying a union bound over failure probabilities, we see that with constant probability, we have

$$\sum_{i=1}^n \|\tilde{U}_{i,:}^k P^{(i)} - A_{i,:} D_{W_{i,:}} S'\|^2 + \lambda\|\tilde{U}_{i,:}^k\|^2 \leq (1 + \varepsilon)\text{OPT} + \tau.$$

Finally, since the degree of the polynomial system using $\tilde{U}^k$ is simply $O(k)$, our theorem follows. $\qquad\square$

# E    Proof of Theorem 9

*Proof.* For all $i$, we want to show there exists some matrix $B$ such that $\eta(P^{(i)}(P^{(i)})^T + \lambda I_k) \preceq B \preceq P^{(i)}(P^{(i)})^T + \lambda I_k$ with constant probability. Then, we may simply guess $B^{-1}$ with only an additional $O((s + \log(1/\varepsilon))^2)$ variables and apply Theorem 7, express a $k$-order approximation of $\tilde{U}_{i,:}$ as

$$\tilde{U}_{i,:}^k = A_{i,:}D_{W_{i,:}}S'(P^{(i)})^T p_k(P^{(i)}(P^{(i)})^T + \lambda I_k, B^{-1})$$

where $p_k$ is a degree $O(k)$ polynomial that approximate the inverse and apply the same analysis as Theorem 8 to see that $k = O(\eta^{-1}\log(c_B/\tau))$ suffices.

In fact, we will explicitly construct $B$. Let $D \in \mathbb{R}^{n \times n}$ be the diagonal matrix with diagonal entries $l_W$. Let $P = V^*DS = RS$, where $R = V^*D_l$. Also, we define $P^{(i)} = V^*D_{w_i,:}S = R^{(i)}S$. Then, we see that by our bounds on $W$,

$$\frac{l_W}{u_W}R^{(i)}(R^{(i)})^T \preceq RR^T \preceq R^{(i)}(R^{(i)})^T$$

By using similar arguments for condition number bounds as in Claim A.3 in Theorem 2, we see that

$$\frac{l_W}{u_W \log(n)}R^{(i)}SS^T(R^{(i)})^T \preceq RSS^TR^T \preceq \log(n)R^{(i)}SS^T(R^{(i)})^T$$

with high probability. So, we set $B = (1/\log(n))RSS^TR$ and that implies that $\eta = 1/\log(n)^2(u_W/l_W)$. Then, using Theorem 7, we conclude with the same analysis as in Theorem 8. □

# F    A Note on the Experiments

It may surprise the reader to see that the objective values when using sketching slightly outperform the objective values without using sketching and that the objective value improves as the sketching dimension decreases. In theory, this should not happen because in low rank approximation problems, it never hurts the objective value to increase the number of columns of $U$ and number of rows of $V$ since one can simply add 0's.

This phenomenon arises due to the use of the alternating minimization heuristic. Although an ideal low rank approximation algorithm would recognize that if $U$ and $V$ have more columns / rows, then one only needs to add 0's, we found in our experiments that alternating minimization tended to add mass to those extra columns / rows. This extra mass resulted in a higher contribution from the regularization terms. Thus, by sketching onto fewer dimensions, the alternating minimization heuristic was improved because it couldn't add mass in the form of extraneous columns for $U$ or rows for $V$.