[Reviews · NeurIPS 2019]

Reviewer 1



I found the results of the article useful but difficult to follow without being an expert on previous results on weighted low-rank approximation. Additionally, I think the clarity of the paper could be highly improved. For example, at the end of the Introduction section, they refer to a matrix A which is not defined before, making the reading difficult. Similarly, in line 89, matrices D_{W_{i,:}} and D_{W_{:,j}} are used but defined later in lines 93-94. Also, in line 121 the concept of non-negative rank is used but not defined. Experimental results are rather weak. For example, the particular case of matrix completion is not covered, which consists of using a binary weight matrix. The regularization parameter lambda was set to 1 without a clear justification.

Reviewer 2



- Originality: - The authors propose to look at provable guarantees for regularized weighted low rank approximation, while the related works mainly study the weighted low rank approximation problem. Thus, the problem in this paper is new. However, the proposed algorithm and the techniques to analyze its complexity is similar to those in [RSW16], which makes the contribution of this paper incremental. - The related work is adequately cited. - Quality: - The authors provide theoretical analysis of their claims in the supplementary material. - Though the authors claim that the proposed algorithm can be significantly faster than other algorithms, they did not conduct experiments to support the claim. The authors may want to conduct experiments to show how much faster will the proposed algorithm be than the previous algorithm (e.g., SVD). - Clarity: This paper is clearly written and well organized. It provides enough information to reproduce its results. - Significance: In this paper, the authors theoretically show that we can solve the regularized weighted low rank approximation problem with the time complexity bounds in terms of the statistical dimension rather than the rank. In many real applications, the statistical dimension is lower than the rank. The derived complexity bounds ensure that we can solve regularized weighted low rank approximation problems fast. Thus, the result of this paper is important. [RSW16] Ilya 354 P. Razenshteyn, Zhao Song, and David P. Woodruff. Weighted low rank approximations with provable guarantees. In Proceedings of the 48th Annual ACM SIGACT Symposium on Theory of Computing, STOC 2016, Cambridge, MA, USA, June 18-21, 2016, pages 250–263, 2016.

Reviewer 3



Thank you for addressing my review comments. I am somewhat satisfied with the answers. I will increase the score provided that authors will add the suggested experimental comparisons in the rebuttal to the final version of the paper. -------------------------------------------------- This paper discusses improvement for weighted low-rank for matrix approximation. This paper closely follows methodologies by [Razenshteyn et al. ’16]. Overall the paper is well-written, however, though the theoretical analysis is strong enough to show an improvement over [Razenshteyn et al. ’16], the paper does not properly demonstrate the results through experiments. One of the main contributions they claim is the improvement of the running time of matrix approximation (in lines 87-90), however, there is no empirical evidence in the paper ([Razenshteyn et al. ’16] gives experiments on running time). Can the authors demonstrate this speed up by experiments? Experiments: Why do you set the regularization parameter \lambda equal to 1? Why not cross-validate? Any theoretical justification? I feel that the dataset sizes are too small (dimensions of 1000) to demonstrate that the proposed method is useful in real-world applications. Can the authors provide at least one experiment with a large dataset? As I mentioned earlier, it is desirable to see experiments with running times.

[Author Response · NeurIPS 2019]

We thank the reviewers for their comments.

**Reviewer 1.** In the final draft, we will edit the exposition to make it friendlier to non-expert readers. Specifically,
we will correct the reference to matrix $A$ at the end of the introduction (line 68) and change it to matrix $M$. We will
also move the definitions of $D_{W_{i,:}}$ and $D_{W_{:,j}}$ to a point before their first use in line 89. We will provide the standard
definition for *non-negative rank*.

Like [RSW16], our setting involves a more general family of weight matrices $W$ than just binary matrices. Since
our proofs dealt with weight matrices that were not necessarily binary matrices, we wanted our experiments to use
non-binary weight matrices to highlight the fact that we weren't just studying matrix completion. We have additional
experiments involving the NIPS and synthetic datasets that use binary weight matrices which turn out similarly to our
current experiments. For the final draft, we would be happy to add these experiments to the appendix, as well as some
additional experiments with varying regularization parameter values.

**Reviewer 2.** We will address speed and SVD-related issues in the comments to Reviewer 3.

We believe the contribution of this work over [RSW16] is more than incremental because even though the algorithmic
steps may be similar, the proof techniques required are quite different. Most provable sketching results for Low Rank
Approximation (LRA) problems do not have sketch sizes that can be significantly smaller than the rank. The small
sketch size means imitating the analysis of [RSW16] is insufficient because when one solves a regression problem on a
matrix with fewer rows than columns one always gets 0. Furthermore, the proof was achieved without the incoherence
assumptions on the input matrix that are popular in the matrix completion literature. Thus, we needed a finer analysis
based on condition numbers and tail bounds on the singular vectors because directly following the approach of [RSW16]
will fail to give sharp enough inequalities. We elaborate on this in lines 101 to 108.

We also improve the results of [RSW16] by providing fast $2^{\mathrm{poly}(r \cdot sd)}$ algorithms (as opposed to $n^{\mathrm{poly}(r \cdot sd)}$) in the case
when the ratio of the largest to smallest entries of the weight matrix is controlled and the largest singular value of our
regression matrices is small relative to lambda. We achieve this by replacing the Cramer's rule-based approach in
[RSW16] with different techniques from optimization like Richardson's Iteration.

**Reviewer 3.** The primary focus of our work is on the theoretical side rather than the experimental side. We would like
to reiterate that our algorithm has a greatly improved running time when compared to other $(1 + \epsilon)$-approximation
algorithms for our regularized, weighted setting. The theoretical running time is not being compared to that of singular
value decomposition because SVD is not a $(1 + \epsilon)$-approximation algorithm for the regularized, weighted setting.

We included SVD in our experiments because it is widely used in practice for LRA type problems and to demonstrate
that it results in high objective values for the loss function. Given the significantly higher objective values and the fact
that SVD does not provide a $(1 + \epsilon)$-approximation algorithm for our problem (because it is NP-complete) we did
not think it was appropriate to compare its speed with our algorithm. However, we did think it was fair to compare
the sketched version of our algorithm to an *unsketched* version of our algorithm. As described in line 299, we ran
experiments that showed that alternating minimization with sketching was between 1.43 and 2 times as fast as alternating
minimization without sketching. We can add a table to the final draft.

We also wanted to emphasize in line 258 that the purpose of the experiments was to show that even if one sketches
down to the statistical dimension, which can potentially be much lower than the rank of the matrix, it is possible to do
this without blowing up the objective value in regularized weighted LRA. While the focus on the theoretical side of the
paper was on the running time, the focus on the experimental side of the paper was on the dimension reduction.

This is because this algorithm and the algorithm in [RSW16] which we improve on both use polynomial system solvers
which are costly. In fact, the implementation in [RSW16] could barely handle target ranks and sketching dimensions
larger than 2 and matrix dimensions larger than 100. Our experiments involve target ranks of at least 50 and matrix
dimensions in the 1000s. Thus, we feel that the dataset sizes show a marked improvement over the prior work but we
can include an even larger dataset for the final draft.

Although they are costly, polynomial system solvers do have provable theoretical guarantees which is why we invoked
them in the theoretical part of our paper. In practice, heuristics like alternating minimization are often faster but they
lack provable theoretical guarantees without making assumptions on the input, which we do not. Since our experiments
were not using the polynomial system solver technique described in the theoretical sections of our papers but they were
using the same sketches, we decided to focus the experiments on dimension reduction.

We obtained our $\lambda$ value by hand-tuning and felt that it was in a sweet spot that avoided underfitting and overfitting. We
can add experiments with varying regularization parameter values, or $\lambda$ values tuned by cross-validation, in the final
draft.

[Meta-Review · NeurIPS 2019]

This paper studies weighted low-rank for matrix approximation, providing theoretical improvements over the existing results by [Razenshteyn et al. ’16]. The authors should include more experimental results in the final version of the paper as suggested in the rebuttal.